# Epidemiology and Future Burden of Vertebral Fractures: Insights from the Global Burden of Disease 1990–2021

**DOI:** 10.3390/healthcare13151774

**Published:** 2025-07-22

**Authors:** Youngoh Bae, Minyoung Kim, Woonyoung Jeong, Suho Jang, Seung Won Lee

**Affiliations:** 1Department of Neurosurgery, Korean Armed Forces Capital Hospital, Seongnam 13487, Republic of Korea; yobae05@gmail.com; 2Department of Precision Medicine, Sungkyunkwan University School of Medicine, Suwon 16419, Republic of Korea; saviour_sh@icloud.com; 3Department of MetaBioBealth, Sungkyunkwan University School of Medicine, Suwon 16419, Republic of Korea; dbssus123@gmail.com (M.K.); hbm06014@naver.com (W.J.); 4Department of Family Medicine, Kangbuk Samsung Hospital, Sungkyunkwan University School of Medicine, Seoul 03181, Republic of Korea

**Keywords:** vertebral fracture, global burden of disease, prevalence, YLDs, 2050 projections

## Abstract

**Background/Objectives**: Vertebral fractures (VFs) are a global health issue caused by traumatic or pathological factors that compromise spinal integrity. The burden of VFs is increasing, particularly in older adults. **Methods**: Data from the Global Burden of Disease 2021 were analyzed to estimate the prevalence, mortality, and years lived with disability due to VFs from 1990 to 2021. Estimates were stratified according to age, sex, and region. Bayesian meta-regression models were used to generate age-standardized rates, and projections for 2050 were calculated using demographic trends and the sociodemographic index. Das Gupta’s decomposition assessed the relative contributions of population growth, aging, and prevalence changes to future case numbers. **Results**: In 2021, approximately 5.37 million people (95% Uncertainty Interval [UI]: 4.70–6.20 million) experienced VFs globally, with an age-standardized prevalence of 65 per 100,000. Although the rates have declined slightly since 1990, the absolute burden has increased owing to population aging. VF prevalence was the highest in Eastern and Western Europe and in high-income regions. Males had higher VF rates until 70 years of age, after which females surpassed them, reflecting postmenopausal osteoporosis. Falls and road injuries were the leading causes of VF. By 2050, the number of VF cases is expected to increase to 8.01 million (95% UI: 6.57–8.64 million). **Conclusions**: While the age-standardized VF rates have decreased slightly, the global burden continues to increase. Targeted strategies for the early diagnosis, osteoporosis management, and fall prevention are necessary to reduce the impact of VFs.

## 1. Introduction

Vertebral fractures (VFs) are structural injuries of the spine that result from trauma or pathological conditions and represent a major global health concern [1]. While commonly associated with high-energy impacts like traffic accidents, falls, and sports injuries, VFs may also result from aging, poor nutrition, endocrine disorders, metabolic disorders, and malignancies [2]. Younger adults are more likely to experience traumatic VFs primarily during occupational or athletic activities. Conversely, even low-energy trauma may cause fractures in older adults, emphasizing the need for age-specific preventive strategies.

VFs are associated with acute pain, functional impairment, and hospitalization and may lead to long-term consequences such as chronic pain, mobility limitations, and decreased quality of life [3]. These outcomes affect independence, especially among older adults, whose increased fall risk further elevates fracture risk [4]. Moreover, long-term disability imposes financial and emotional strain on caregivers [5] and contributes significantly to rising healthcare expenditures owing to repeated interventions, rehabilitation, and medication needs [2]. Globally, VFs account for billions of dollars in annual direct and indirect costs, and their burden continues to grow [6], particularly in low- and middle-income countries (LMICs), where underdiagnosis is prevalent [7].

Recent Global Burden of Disease (GBD) analyses have provided essential baseline estimates for the prevalence and disability burden of VFs [8,9,10], but they remain largely descriptive and have several limitations. They do not offer forward-looking projections beyond the observed period and focus primarily on fall-related fractures, neglecting other major causes such as road traffic injuries and cancer-related fractures. Additionally, prior studies report only aggregate estimates without disaggregating the future burden by etiology, age group, region, or Socio-demographic Index (SDI). To address these gaps, we utilized the latest fully validated GBD 2021 dataset to produce the first etiology-specific, long-term forecasts through 2050 of VF incidence, prevalence, and years lived with disability (YLDs). Our projections are stratified by age, region, and the SDI quintile, with particular attention to postmenopausal women aged ≥ 65 years, who are expected to experience the steepest increases. Furthermore, we performed region-specific Das Gupta decomposition analyses to quantify the relative contributions of population growth, population aging, and changes in age-specific rates to the projected burden.

By combining comprehensive forecasting with decomposition analysis, our study provides a more nuanced and policy-relevant understanding of the evolving global burden of VFs. Thus, this study provides an essential framework for future prevention, early detection, and health policy planning efforts globally.

## 2. Materials and Methods

### 2.1. Study Overview

This study used data from GBD 2021 to estimate the global prevalence, mortality, and YLDs associated with VFs stratified by age and sex. In addition, the prevalence of VFs up to 2050 was projected. All analyses followed the GBD 2021 and GATHER protocols [11] with standard handling of missing or incomplete data.

### 2.2. Case Definition

VFs are structural disruptions of the spine caused by traumatic or pathological factors that result in functional impairments. This study focused on VFs caused by traumatic events, like road traffic accidents, falls, and sports injuries, as well as non-traumatic pathological causes, including osteoporosis, malignancies, and metabolic or degenerative diseases. In the GBD framework, VFs are identified using ICD-9 and ICD-10 codes recorded in hospital, outpatient, survey, and insurance datasets.

The analyses were based on GBD 2021 Data Input Sources, available from the Global Health Data Exchange (GHDx) website (https://ghdx.healthdata.org/gbd-2021/data-input-sources, accessed on 14 January 2025). VF prevalence and YLD estimates from 1990 to 2021 were derived by synthesizing data from population-based studies, global and national health surveys, and datasets provided by international collaborators. The dataset spans diverse demographic and etiological categories relevant to VFs. Each source was assigned a unique identifier and cataloged within the GHDx system. The authors accessed and reviewed the estimates as part of the GBD modeling framework.

### 2.3. Modeling and Data Processing

Bayesian meta-regression was conducted using the ‘brms’ R package version 2.21.2 for Bayesian estimation, which allowed for hierarchical modeling with random effects and the inclusion of covariates [12]. This model was used to estimate the prevalence of VF and YLDs according to age, sex, location, and year. Data sources lacking detailed information on age or sex were disaggregated proportionally before analysis. For sources with nonstandard case definitions, adjustments were made using network meta-regression to align the estimates with the reference case definitions [13].

Before modeling, data reported across broad age groups or combined for both sexes were disaggregated by age and sex. This process followed methods described in previous studies [14]. This method was used to adjust for sex-specific estimates and correct for biases in heterogeneous data sources [15]. MR-BRT is a meta-regression tool developed within the GBD framework to adjust for bias and heterogeneity in data by trimming outliers, applying regularization, and incorporating relevant covariates. The female-to-male ratio was estimated at 1.19 (95% uncertainty interval [UI]: 1.03–1.40). For data reported in wide age groups (≥25 years difference), age-specific estimates were generated by applying the age pattern of VFs, derived from the GBD 2021, to disaggregate the data into 5-year age intervals.

Data sources reporting on VFs using alternative case definitions were adjusted to align with the reference definitions. Bias was adjusted using the MR-BRT tool. By matching data across age, sex, year, and location, MR-BRT network meta-regression was used to estimate the logit differences in prevalence between the alternative and reference definitions, from which adjustment factors were derived [16]. After adjustment, any data point with an age-standardized median absolute deviation >1.5 times higher than the sex- and location-specific mean prevalence was considered an outlier and excluded from the analysis [17]. This 1.5-fold threshold was selected based on the general statistical principle that such values typically lie beyond three standard deviations from the mean. These steps ensured consistency across diverse data sources and enabled a reliable estimation of global VF prevalence by age, sex, and location.

Prevalence estimates were generated using the Bayesian meta-regression tool DisMod-MR 2.1.1 and stratified according to age, sex, location, and year [18]. Individuals <5 years old were assumed to have no VFs. UIs were calculated using the 2.5th and 97.5th percentiles of 1000 posterior draws following model convergence [19]. Estimates were reported at the global, super-regional (seven regions), regional (21 regions), and national levels. The GBD framework quantifies the impacts of diseases and injuries in terms of prevalence, incidence, and mortality [20].

The final model used the following equation:(1)$$Logit\;\left(predicted\;prevalence\right)=\;\beta_{1}\left(SDI\right)+\;\alpha_{l,a,s}$$
where *β*_1_ represents the fixed coefficient of the SDI, and α denotes random intercepts for location (*l*), age group (*a*), and sex (*s*). Future values were calibrated against 2020 baseline estimates from GBD 2021. To account for temporal trends, prevalence estimates from the GBD 2021 for 2020 were used as the baseline, with adjustments applied to all future projections through 2050. Covariates used in the DisMod-MR model included age, sex, year, region, and the SDI. The SDI was used both as a stratifying and predictive variable in forecasting future VF burdens.

### 2.4. Decomposition Analysis

Das Gupta decomposition analysis was conducted to assess the relative contributions of population growth, population aging, and changes in age-specific prevalence to the projected increase in VF cases from 2020 to 2050 [21]. Validation tests were performed using estimates from 1990 to 2020 to predict prevalence between 2030 and 2050. These projections were used to calculate the disability burden in terms of the YLDs. This approach allowed us to isolate the impacts of demographic shifts and epidemiological changes on the projected VF burden.

### 2.5. Risk Estimation

In the GBD 2021, key injury-related risk factors for VFs included falls, road injuries, other transport injuries, and violence-related causes such as police conflict and execution. VFs were primarily caused by external trauma such as traffic accidents, falls, and occupational injuries. Low-energy trauma, particularly in combination with osteoporosis, can lead to fractures in older adults. These risk factors were incorporated into the DisMod-MR 2.1 model as covariates to improve predictive accuracy, particularly in regions with sparse or low-quality data.

### 2.6. Projection of Estimates

Projections of VF prevalence through 2050 were based on age-, sex-, and location-specific estimates from the GBD 2021 for 1990–2021. A regression model incorporating the SDI as a predictor was used to estimate future prevalence. The predicted rates were adjusted using population forecasts and multiplied by the projected population size to derive the number of future cases. Future values were calibrated as described in Section 2.4. In addition, Das–Gupta decomposition was applied to quantify the relative influence of demographic factors, including population growth and aging, and changes in the prevalence of the total projected case burden between 2020 and 2050. The future overall prevalence was estimated by applying region- and age-specific prevalence trends, along with regional and SDI-based projections.

## 3. Results

This study evaluated the global, regional, and national burdens of VFs from 1990 to 2021 (Figure 1). This figure shows that high-prevalence regions were North America, Western and Eastern Europe, and Australasia, while low-prevalence regions included Andean Latin America and East Asia.

### 3.1. Global and Regional Prevalence of VFs in 2021

In 2021, the estimated number of individuals who experienced VFs globally was 5,371,438 (95% UI: 4,703,837–6,196,132). The global age-standardized prevalence rate was 65 per 100,000 people (95% UI: 57–75), reflecting a slight decline of −0.21% (95% UI: −0.22% to −0.19%) from 1990. The number of YLDs attributable to VFs was 545,923 (95% UI: 366,571–757,099), with a similar age-standardized decrease of −0.21% over the same period.

As depicted in Table 1, Australasia recorded the highest age-standardized prevalence rate (182 per 100,000; 95% UI: 158–209), followed by high-income North America (158 per 100,000; 95% UI: 138–178) and Western Europe (151 per 100,000; 95% UI: 132–172). Most regions experienced a modest decrease in age-standardized prevalence from 1990 to 2021, with the greatest reduction observed in Central Europe (−0.24%; 95% UI: −0.26–0.22). However, certain regions, such as the Caribbean, Oceania, and East Asia, showed slight increases in both prevalence and YLDs over the same period.

### 3.2. Statistical Significance of Regional Trends

Most regions experienced statistically significant changes in age-standardized prevalence rates between 1990 and 2021 (Table 1). Central Europe showed the greatest reduction (0.24%, 95% UI: −0.26 to −0.22), followed by Southeast Asia (0.10%, 95% UI: −0.15 to −0.04) and Southern Sub-Saharan Africa (0.29%, 95% UI: −0.32 to −0.25). In contrast, several regions demonstrated significant increases, including the Caribbean (+0.24%, 95% UI: +0.14 to +0.40), Oceania (+0.21%, 95% UI: +0.11 to +0.31), and East Asia (+0.13%, 95% UI: +0.09 to +0.18), as their 95% UIs did not include zero. These findings indicate heterogeneous trends across regions with a mix of increasing and decreasing global VF burdens.

### 3.3. Differences by SDI

As shown in Table 1, the burden of VFs varied markedly according to sociodemographic characteristics. Regions with high SDI, including Western Europe (151 per 100,000) and high-income North America (158 per 100,000), exhibited the highest age-standardized prevalence rates in 2021. In contrast, sub-Saharan Africa reported the lowest burden, with rates ranging from 23 to 26 per 100,000 individuals across the subregions. Interestingly, although the prevalence remained relatively stable in the low- and middle-SDI regions, high-SDI regions experienced a modest but statistically significant decline. These patterns likely reflect the differences in the aging population, diagnostic capacity, and injury risk profiles across income levels.

### 3.4. National Level Prevalence in 2021

At the national level, in 2021, the highest age-standardized prevalence of VFs was observed in Andorra (241 per 100,000; 95% UI: 207.3–276.9), followed by Belgium (210 per 100,000; 95% UI: 183.4–241.2). The largest decreases in age-standardized prevalence between 1990 and 2020 were seen in Portugal (74.7 per 100,000; 95% UI: 67.6–82.8) and Switzerland (83.9 per 100,000; 95% UI: 73.1–92.8).

### 3.5. Prevalence by Age, Sex, and Cause

As shown in Figure 2, the prevalence of vertebral fractures (VFs) increased sharply after the age of 60, with a more pronounced rise in females. The prevalence in men was higher than that in women up to the age of 70, after which that in females surpassed it. The overall difference in age-standardized prevalence between sexes was small, with males showing a slightly higher rate (−10.02 per 100,000; 95% UI: −8.13 to 28.67). In 2021, the leading global causes of VFs were falls and road injuries, accounting for approximately 1% of the total YLDs.

### 3.6. Projection of VF Burdens Through 2050

As shown in Table 2, while the global age-standardized prevalence rate is projected to remain stable at approximately 0.08% (95% UI: 0.07–0.09) from 2030 to 2050, the absolute number of individuals living with vertebral fractures is expected to increase from 6.89 million (5.82–7.63) in 2030 to 8.01 million (6.57–8.64) by 2050, primarily due to population aging. East Asia is projected to experience the largest absolute increase, with the number of cases rising from 2.86 million (2.18–3.56) in 2030 to 9.97 million (6.92–13.13) in 2050, alongside a marked rise in prevalence rate from 0.19% to 0.74%. Similarly, North Africa and the Middle East are expected to see substantial increases, with cases rising from 0.78 million (0.53–1.39) in 2030 to 1.86 million (1.10–4.36) by 2050, and prevalence increasing from 0.11% to 0.21%. In contrast, high-income Asia Pacific is the only region projected to show a consistent decline in both prevalence and case numbers, with prevalence decreasing from 0.06% in 2030 to 0.03% in 2050, and total cases dropping from 0.10 million (0.09–0.11) to 0.04 million (0.04–0.05).

As shown in Figure 3, Between 2020 and 2050, the global number of individuals living with vertebral fractures is projected to increase substantially, rising from approximately 17.7 million in 2020 to 38.2 million in 2050, more than doubling over three decades. Eastern Europe and North Africa & the Middle East showed the largest increases (>150%), driven mainly by population aging and rising prevalence rates, respectively. High-income Asia Pacific shows a net decrease (~−40%), largely due to a strong decline in age-specific prevalence rates, despite population aging. These trends are largely driven by an increase in prevalence rates and population aging, suggesting substantial regional variation in the future burden of VFs. Appendix A presents differences in prevalence by age group (<65, Appendix A vs. ≥65 years, Appendix A) and by country SDI level (Appendix A).

## 4. Discussion

### 4.1. Global Prevalence and Burden

Over the past three decades, the global burden of VFs has increased substantially, largely because of population growth and aging. Between 1990 and 2021, the number of VF cases increased by approximately 38%, and the VF-related YLDs increased by 75%, reaching nearly 550,000 in 2021 (Table 1). In contrast, age-standardized prevalence and YLD rates have shown a modest global decline, with annual percentage changes ranging from −0.1% to −0.3%. This divergence indicates that while more people live with VF-related disabilities, age-specific risks have likely decreased due to improved outcomes. However, the growing number of affected individuals underscores the need for more robust prevention and management strategies.

In 2021, the total YLDs for all fracture types exceeded 25.8 million, with VFs accounting for a modest but meaningful share, particularly among older adults. Among musculoskeletal conditions, lower back pain remains the leading contributor to disability, responsible for an estimated 60–70 million YLDs globally. As life expectancy increases, the health system burden attributable to VFs is expected to increase unless more effective interventions are implemented.

Consistent with previous research, this study emphasized the urgent need for policy responses to address the growing VF burden associated with population aging [8,9]. However, unlike earlier studies that focused exclusively on fall-related vertebral body fractures [9] and relied on GBD 2019 data [8,9], this analysis incorporated a broader set of etiologies, including falls, road traffic injuries, and malignancies, using the updated GBD 2021 dataset. By applying MR-BRT and the Das Gupta decomposition, we assessed historical trends from 1990 to 2021 and projected the burden through 2050. Our findings indicate that with accelerating population aging, VFs are becoming increasingly diverse, with a growing proportion attributable to osteoporosis and other pathological conditions.

### 4.2. Socioeconomic Impact of VFs

VFs impose considerable socioeconomic burdens on individuals, healthcare systems, and society [22]. Patients with VFs frequently experience chronic pain, reduced mobility, and spinal deformities that significantly impair their daily functioning and work capacity [23]. These consequences translate into substantial direct medical costs, particularly for acute treatment and rehabilitation, and indirect costs owing to productivity loss. The financial burden is particularly pronounced in high-income countries (HICs). In the United States, approximately 70,000 VF-related hospitalizations occur annually, with acute care costs exceeding USD 500 million per year [24]. Similarly, the European Union incurs an estimated EUR 377 million annually in hospital care for patients with VFs, with VF-related admissions accounting for nearly 63% of the cost of hip fractures [25].

In addition to direct medical expenses, families of patients with severe VF-related disabilities often bear substantial financial and emotional burdens [26]. In middle-aged adults, VF-associated disabilities can result in a prolonged absence from work or early retirement, leading to significant income loss. Among older adults, long-term care at home or admission to nursing facilities may become necessary, further increasing household expenditure. Thus, beyond clinical care, VFs impose long-term financial costs and significantly diminish the quality of life for patients and their families [27]. These effects highlight the need for integrated medical and social interventions.

### 4.3. Disparities Between HICs and LMICs

Striking disparities in the burden of VFs exist between HICs and LMICs [28]. Paradoxically, age-standardized VF incidence and disability burden are often higher in HICs than in less-developed regions. GBD data suggest a positive correlation between a country’s SDI, which incorporates income, education, and fertility rates, with both VF prevalence and YLDs. For instance, North America, Western Europe, and East Asia report some of the highest VF burdens globally.

Several factors could explain this pattern. HICs typically have older populations with longer life expectancies, resulting in a larger proportion of individuals at risk for osteoporotic fractures. Additionally, robust surveillance systems and greater access to diagnostic imaging in HICs are likely to result in higher detection and reporting rates [29]. In contrast, many VF cases in LMICs may go undiagnosed or unreported owing to limited healthcare infrastructure. Lifestyle factors in affluent societies, such as physical inactivity and higher rates of obesity, may also contribute to an increased fracture risk [23]. LMICs generally have a younger population structure with fewer older adults vulnerable to osteoporosis-related fractures. However, as LMICs undergo demographic transition and economic growth, their VF burden is expected to significantly increase [30]. Countries in Asia and Latin America, in particular, are experiencing rapid population aging and lifestyle changes, with evidence of increasing fracture incidence already emerging [29]. Compounding this issue, limited access to osteoporosis screening and treatment in low-income settings may result in a persistently high prevalence of preventable fractures [31].

These disparities also extend to treatment outcomes. Although HICs invest in post-fracture care and rehabilitation, LMICs often lack resources for surgical intervention and long-term management, potentially leading to a greater disability burden [29]. HICs should prioritize implementing fragility fracture liaison services to reduce recurrent VFs. In LMICs, integrating osteoporosis screening into primary care and subsidizing calcium/vitamin D supplements could be cost-effective first steps.

### 4.4. Age-Related VF Burden

VF burden increases markedly with age. The incidence of VF remains relatively low in adults aged <60 years. In contrast, individuals aged ≥60 experience a sharp rise in VF occurrence, primarily owing to fragility fractures associated with osteoporosis and minor falls. Epidemiological data indicate an exponential increase in both the VF incidence and disability burden with age. As of 2021, the VF burden will peak in adults aged 60–80 years. (Table 1).

Age-related physiological changes, such as declining bone mineral density and increased risk of falls, contribute to both the frequency and severity of VFs in older populations. Postmenopausal women and older men are particularly vulnerable to vertebral compression fractures following low-energy trauma, a condition rarely observed in younger adults [32]. One study found that individuals aged ≥ 70 accounted for nearly half of all VF cases, underscoring the significant role of aging in fracture susceptibility [33].

Aging of the global population is becoming a major driver of VF burden. In rapidly aging countries such as Japan and Italy, the demand for healthcare and social services related to osteoporotic fractures is steadily increasing. Even in regions where age-specific VF incidence may decline, the growing proportion of older adults is expected to lead to a continued increase in the absolute VF burden in the coming decades.

### 4.5. Sex Differences in VF Burden

Globally, the age-standardized rates of VF incidence and YLDs are higher in men than in women, indicating a greater adjusted risk of VF occurrence and disability among males (Table 1). This disparity may be attributed to a greater exposure to traumatic risk factors such as occupational injuries, traffic accidents, and high-impact sports during adolescence and middle adulthood in men [34]. Additionally, osteoporotic fractures tend to be under-recognized in men, and when they occur, they may be more severe or sudden in onset [30].

However, this pattern is reversed after the age of 65 years, with an increasing VF incidence in women [35]. Postmenopausal estrogen deficiency accelerates bone loss in women, increasing the risk of VFs even after low-energy trauma [36]. According to an international study, one in four women over the age of 50 years is expected to experience VF during her lifetime, which is several times greater than the risk observed in age-matched men [37]. Furthermore, women generally live longer than men, leading to a higher cumulative VF burden among older females who survive in high-risk age brackets [38]. These findings emphasize the need for sex-specific prevention strategies that reflect differences in risk exposure across the life course and address the distinct biological and social determinants that influence VF burden in men and women.

### 4.6. Shifting VF Burden in Women by Age and SDI

Appendix A and Appendix A illustrate trends in the prevalence and YLDs due to VFs among women aged ≤65 and ≥65 years, respectively, across global and regional levels. While the global prevalence and YLDs have declined slightly (−0.21%), both metrics have increased substantially among women aged ≥65 years. Appendix A illustrates how the VF-related burden increased by 17.4% in high-SDI countries and by 28.1% in low- to middle-SDI countries, highlighting the growing impact of population aging on the VF burden. These findings underscore the need for targeted prevention strategies in older women, including early osteoporosis screening and fall prevention strategies. Moreover, approaches differentiated by SDI levels are warranted, focusing on quality improvement in high SDI settings and infrastructure development in lower SDI contexts.

### 4.7. Risk Factors for VFs

VFs result from a combination of direct causes, such as falls and road injuries, and from underlying risk factors that increase an individual’s vulnerability to trauma or bone fragility [39]. Globally, the most common precipitating event in VFs is trauma owing to falling. In older adults, even low-energy falls, such as slipping from a standing height, can lead to fractures in already weakened vertebrae [40]. Most fractures in older adults are fragility fractures, often exacerbated by underlying osteoporosis, and may occur not only from falls but also during routine daily activities [41].

Several contextual factors influence VF prevalence across regions, including dietary habits, physical activity levels, and frequency of trauma exposure. For example, East Asian countries undergoing rapid population aging, traditionally characterized by a low calcium intake, may face a growing VF burden in the coming years. In contrast, some European countries have implemented effective strategies for early osteoporosis screening and fall prevention, which may contribute to the stabilization or reduction in VF incidence [42].

### 4.8. Implications for Policy and Practice

Figure 3 illustrates the decomposition analysis highlights that the drivers of VF burden differ by region. In high-income regions, such as Western Europe and Australasia, population aging was the predominant factor, suggesting the need for early osteoporosis screening, fall prevention, and geriatric care pathways. In contrast, the increasing burden in sub-Saharan Africa and South Asia is largely due to population growth, which underscores the importance of expanding the primary care infrastructure and implementing community-based education and surveillance programs. Regions with declining age-standardized rates, such as Eastern Europe, may provide models for effective interventions. These findings suggest that early screening programs, fall prevention strategies, and broader osteoporosis management initiatives—particularly tailored to aging populations and high-risk SDI groups—will be essential to mitigate future VF burdens.

### 4.9. Strengths and Limitations

This study leveraged the GBD framework, incorporating data from 204 countries between 1990 and 2021, enabling robust temporal and geographic comparisons. The use of large-scale datasets and systematic modeling approaches, such as DisMod-MR meta-regression, allows for the integration of diverse data sources while accounting for known biases, thereby enhancing the reliability of the estimates. Moreover, including the incidence, prevalence, and YLDs allowed for a comprehensive quantification of the health impact of VFs. Analyses stratified by age, sex, region, and SDI further enabled the identification of high-risk subpopulations.

However, this study had several limitations. First, the quality and availability of primary data (e.g., hospital records and survey data) vary substantially across countries. In low-income settings, surveillance systems for VFs may be incomplete, increasing the likelihood of under-diagnosis. Second, differences in national coding practices may complicate data interpretation; for instance, some VFs may be recorded under related diagnoses such as low back pain or osteoporosis, leading to underestimation. Third, the DisMod-MR model is based on specific assumptions and is therefore subject to modeling uncertainty. Fourth, advancements in imaging technologies, such as MRI and AI-based diagnostic tools, have potentially increased VF detection rates in recent years. Finally, as with all large-scale epidemiological analyses, ecological fallacy remains a concern. For example, the observed correlation between high SDI and elevated VF burden does not imply that individuals in high-SDI countries are biologically more vulnerable but rather reflects population-level factors such as aging and greater healthcare access.

## 5. Conclusions

Using GBD data, we found that while the absolute number of VFs increased, age-standardized rates slightly declined. Significant differences in VF burden were observed across regions, ages, and sexes. Strengthening osteoporosis prevention, fall reduction strategies, and region-specific interventions is crucial for mitigating the global impact of VFs.

## Figures and Tables

**Figure 1 healthcare-13-01774-f001:**
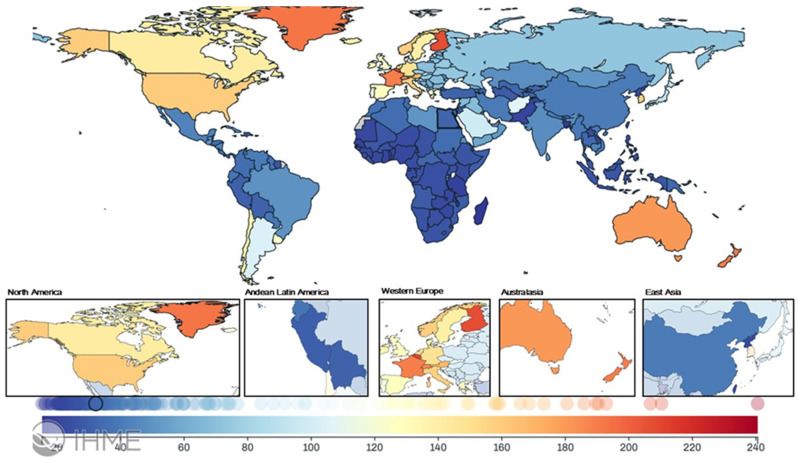
Age-standardized prevalence of the fracture of the vertebral column by country for male and female sexes combined and for all ages in 2021.

**Figure 2 healthcare-13-01774-f002:**
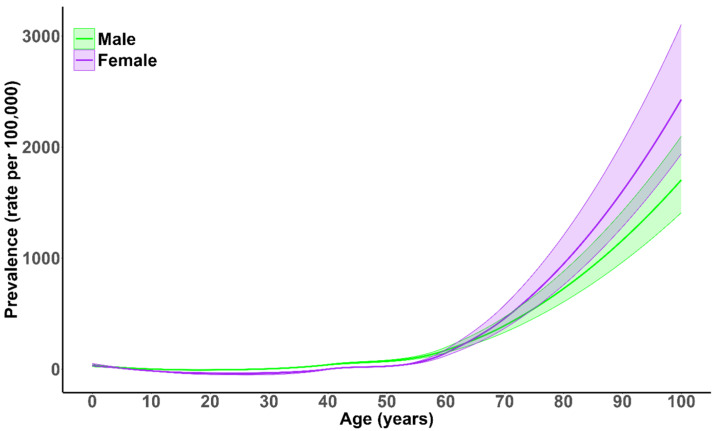
Global prevalence of fracture of the vertebral column by age and sex in 2021. Shaded areas represent 95% uncertainty intervals.

**Figure 3 healthcare-13-01774-f003:**
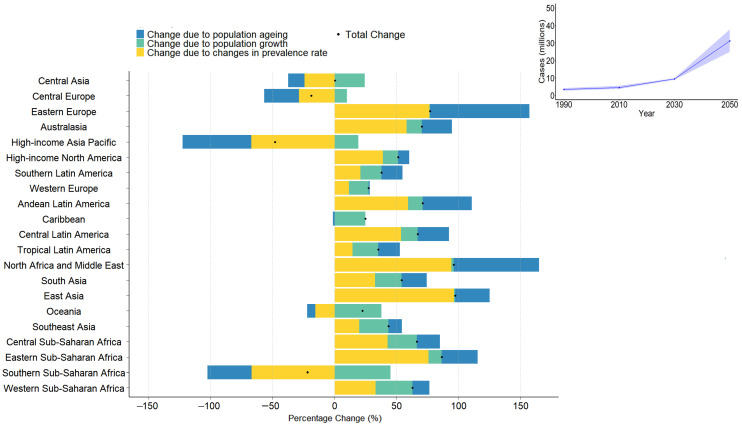
Decomposition of projected change in the number of prevalent vertebral column fracture cases between 2020 and 2050.

**Table 1 healthcare-13-01774-t001:** Global prevalence, YLDs, age-standardized rates of prevalence and YLDs per 100,000 population in 2021, and percentage change between 1990 and 2020 for fracture of the vertebral column, by GBD regions and super-regions.

	Number of Prevalent Cases	Age-Standardized Prevalence Rate per 100,000	Percentage Change in Age-Standardized Prevalence Rate from 1990 to 2021	Number of YLDs	Age-Standardized Rate of YLDs per 100,000	Percentage Change in Age-Standardized Rate of YLDs per 100,000 from 1990 to 2021
Global	5,371,438 (4,703,837 to 6,196,132)’	65 (57 to 75)	−0.21% (−0.22 to −0.19)	545,923 (366,571 to 757,099)	65 (57 to 75)	−0.21% (−0.22 to −0.19)
Central Europe, Eastern Europe, and Central Asia	342,037 (291,142 to 402,579)	66 (54 to 80)	−0.21% (−0.24 to −0.19)	35,433 (23,668 to 49,429)	66 (54 to 80)	−0.21% (−0.24 to −0.19)
Central Asia	34,439 (27,868 to 42,185)	38 (31 to 46)	−0.18% (−0.21 to −0.14)	3667 (2411 to 5236)	38 (31 to 46)	−0.18% (−0.21 to −0.14)
Central Europe	116,959 (100,857 to 137,786)	71 (59 to 87)	−0.24% (−0.26 to −0.22)	11,981 (8062 to 16,770)	71 (59 to 87)	−0.24% (−0.26 to −0.22)
Eastern Europe	190,639 (162,961 to 224,778)	72 (60 to 87)	−0.17% (−0.2 to −0.14)	19,785 (13,361 to 27,689)	72 (60 to 87)	−0.17% (−0.2 to −0.14)
High-income	2,760,887 (2,400,572 to 3,146,904)	143 (126 to 162)	−0.17% (−0.19 to −0.15)	274,301 (182,671 to 379,710)	143 (126 to 162)	−0.17% (−0.19 to −0.15)
Australasia	86,326 (74,702 to 98,843)	182 (158 to 209)	−0.08% (−0.13 to −0.03)	8,616 (5703 to 11,844)	182 (158 to 209)	−0.08% (−0.13 to −0.03)
High-income Asia Pacific	402,262 (355,326 to 451,412)	107 (95 to 121)	−0.3% (−0.33 to −0.28)	40,341 (26,946 to 55,606)	107 (95 to 121)	−0.3% (−0.33 to −0.28)
High-income North America	948,836 (822,350 to 1,080,753)	158 (138 to 178)	−0.02% (−0.07 to 0.03)	93,496 (62,165 to 128,945)	158 (138 to 178)	−0.02% (−0.07 to 0.03)
Southern Latin America	91,462 (82,273 to 102,207)	115 (102 to 129)	−0.01% (−0.06 to 0.04))	9371 (6339 to 12,904)	115 (102 to 129)	−0.01% (−0.06 to 0.04)
Western Europe	1,232,001 (1,066,030 to 1,413,658)	151 (132 to 172)	−0.2% (−0.22 to −0.18)	122,477 (81,287 to 171,373)	151 (132 to 172)	−0.2% (−0.22 to −0.18)
Latin America and Caribbean	266,546 (225,114 to 314,403)	44 (37 to 51)	−0.2% (−0.22 to −0.18)	27,849 (18,806 to 39,218)	44 (37 to 51)	−0.2% (−0.22 to −0.18)
Andean Latin America	20,589 (16,961 to 25,108)	32 (27 to 39)	−0.09% (−0.19 to 0)	2181 (1450 to 3098)	32 (27 to 39)	−0.09% (−0.19 to 0)
Caribbean	21,949 (18,704 to 25,857)	43 (36 to 51)	0.24% (0.14 to 0.4)	2262 (1567 to 3155)	43 (36 to 51)	0.24% (0.14 to 0.4)
Central Latin America	103,659 (87,349 to 124,125)	41 (34 to 49)	−0.3% (−0.32 to −0.27)	10,874 (7365 to 15,275)	41 (34 to 49)	−0.3% (−0.32 to −0.27)
Tropical Latin America	120,349 (101,926 to 142,082)	49 (41 to 58)	−0.18% (−0.21 to −0.14)	12,532 (8310 to 17,771)	49 (41 to 58)	−0.18% (−0.21 to −0.14)
North Africa and Middle East	258,874 (204,786 to 333,164)	46 (37 to 58)	−0.02% (−0.1 to 0.06)	27,310 (18,500 to 39,366)	46 (37 to 58)	−0.02% (−0.1 to 0.06)
South Asia	625,367 (518,858 to 748,472)	42 (36 to 50)	0% (−0.03 to 0.04)	64,644 (43,185 to 92,519)	42 (36 to 50)	0% (−0.03 to 0.04)
Southeast Asia, East Asia, and Oceania	937,489 (803,935 to 1,088,650)	38 (32 to 44)	0.08% (0.03 to 0.11)	97,168 (65,417 to 137,111)	38 (32 to 44)	0.08% (0.03 to 0.11)
East Asia	732,220 (632,696 to 852,696)	40 (34 to 47)	0.13% (0.09 to 0.18)	75,654 (50,869 to 106,460)	40 (34 to 47)	0.13% (0.09 to 0.18)
Oceania	3015 (2451 to 3632)	31 (27 to 37)	0.21% (0.11 to 0.31)	321 (212 to 459)	31 (27 to 37)	0.21% (0.11 to 0.31)
Southeast Asia	202,255 (170,360 to 239,427)	31 (27 to 37)	−0.1% (−0.15 to −0.04)	21,193 (14,577 to 29,974)	31 (27 to 37)	−0.1% (−0.15 to −0.04)
Sub-Saharan Africa	180,236 (141,918 to 228,230)	24 (20 to 30)	−0.15% (−0.25 to −0.08)	19,219 (12,656 to 27,755)	24 (20 to 30)	−0.15% (−0.25 to −0.08)
Central Sub-Saharan Africa	23,216 (18,095 to 29,626)	26 (21 to 32)	0.02% (−0.07 to 0.14)	2477 (1667 to 3610)	26 (21 to 32)	0.02% (−0.07 to 0.14)
Eastern Sub-Saharan Africa	69,295 (52,053 to 95,131)	26 (21 to 34)	−0.24% (−0.4 to −0.12)	7361 (4815 to 10,978)	26 (21 to 34)	−0.24% (−0.4 to −0.12)
Southern Sub-Saharan Africa	18,239 (15,229 to 21,548)	26 (22 to 30)	−0.29% (−0.32 to −0.25)	1925 (1290 to 2677)	26 (22 to 30)	−0.29% (−0.32 to −0.25)
Western Sub-Saharan Africa	69,487 (55,451 to 85,899)	23 (19 to 26)	0.02% (−0.02 to 0.07)	7456 (4895 to 10,680)	23 (19 to 26)	0.02% (−0.02 to 0.07)

Data in parentheses are the 95% uncertainty intervals. Region and super-region numbers do not sum up the global prevalence because of rounding. YLDs, years lived with disability.

**Table 2 healthcare-13-01774-t002:** Age-standardized prevalence and cases of fracture of the vertebral column: projections till 2030, 2040, and 2050, globally and by region, with male and female sexes combined.

	Projected Change in Age- Standardized Prevalence Rate (%)			Cases (Millions)		
	2030	2040	2050	2030	2040	2050
Global	0.08% (0.07–0.09)	0.08% (0.07–0.09)	0.08% (0.07–0.09)	6.89 (5.82–7.63)	7.52 (6.25–8.22)	8.01 (6.57–8.64)
Central Asia	0.03% (0.03–0.04)	0.03% (0.02–0.03)	0.02% (0.02–0.03)	0.03 (0.03–0.04)	0.03 (0.03–0.04)	0.03 (0.03–0.03)
Central Europe	0.05% (0.04–0.06)	0.04% (0.03–0.05)	0.03% (0.03–0.04)	0.06 (0.05–0.07)	0.04 (0.03–0.05)	0.03 (0.03–0.04)
Eastern Europe	0.24% (0.2–0.3)	0.36% (0.29–0.44)	0.52% (0.42–0.65)	0.49 (0.39–0.6)	0.69 (0.55–0.84)	0.96 (0.77–1.19)
Australasia	0.27% (0.22–0.3)	0.3% (0.24–0.33)	0.33% (0.26–0.37)	0.09 (0.07–0.1)	0.11 (0.09–0.12)	0.08 (0.08–0.09)
High-income Asia Pacific	0.06% (0.05–0.06)	0.04% (0.03–0.04)	0.03% (0.02–0.03)	0.1 (0.09–0.11)	0.07 (0.06–0.07)	0.04 (0.04–0.05)
High-income North America	0.11% (0.09–0.12)	0.09% (0.08–0.1)	0.08% (0.06–0.09)	0.43 (0.36–0.48)	0.38 (0.31–0.41)	0.32 (0.25–0.35)
Southern Latin America	0.16% (0.14–0.18)	0.17% (0.15–0.2)	0.19% (0.17–0.22)	0.11 (0.1–0.13)	0.13 (0.12–0.15)	0.15 (0.13–0.17)
Western Europe	0.13% (0.11–0.14)	0.11% (0.09–0.12)	0.09% (0.08–0.1)	0.57 (0.49–0.64)	0.49 (0.42–0.55)	0.41 (0.35–0.46)
Andean Latin America	0.03% (0.03–0.03)	0.03% (0.03–0.03)	0.03% (0.03–0.03)	0.03 (0.02–0.02)	0.03 (0.02–0.02)	0.03 (0.02–0.02)
Caribbean	0.04% (0.03–0.05)	0.04% (0.03–0.05)	0.03% (0.03–0.04)	0.02 (0.01–0.02)	0.02 (0.01–0.02)	0.02 (0.01–0.02)
Central Latin America	0.09% (0.08–0.11)	0.12% (0.11–0.14)	0.15% (0.15–0.17)	0.27 (0.25–0.32)	0.38 (0.35–0.43)	0.51 (0.49–0.58)
Tropical Latin America	0.07% (0.06–0.09)	0.08% (0.06–0.11)	0.09% (0.06–0.12)	0.17 (0.13–0.22)	0.19 (0.14–0.26)	0.21 (0.15–0.29)
North Africa and Middle East	0.11% (0.07–0.19)	0.15% (0.09–0.3)	0.21% (0.12–0.49)	0.78 (0.53–1.39)	1.22 (0.78–2.5)	1.86 (1.1–4.36)
South Asia	0.05% (0.05–0.06)	0.06% (0.05–0.07)	0.07% (0.05–0.08)	1.09 (0.9–1.27)	1.25 (1.03–1.45)	1.4 (1.15–1.61)
East Asia	0.19% (0.15–0.24)	0.38% (0.27–0.48)	0.74% (0.52–0.98)	2.86 (2.18–3.56)	5.42 (3.94–6.94)	9.97 (6.92–13.13)
Oceania	0.03% (0.02–0.03)	0.02% (0.02–0.03)	0.02% (0.02–0.03)	0 (0–0)	0 (0–0.01)	0 (0–0.01)
Southeast Asia	0.04% (0.03–0.05)	0.04% (0.03–0.05)	0.05% (0.04–0.06)	0.29 (0.24–0.36)	0.33 (0.26–0.41)	0.36 (0.28–0.45)
Central Sub-Saharan Africa	0.03% (0.02–0.04)	0.03% (0.03–0.05)	0.04% (0.03–0.06)	0.05 (0.04–0.08)	0.07 (0.05–0.1)	0.09 (0.07–0.14)
Eastern Sub-Saharan Africa	0.03% (0.02–0.05)	0.03% (0.02–0.06)	0.03% (0.02–0.06)	0.17 (0.12–0.28)	0.21 (0.15–0.38)	0.26 (0.18–0.5)
Southern Sub-Saharan Africa	0.02% (0.01–0.02)	0.01% (0.01–0.01)	0.01% (0.01–0.01)	0.01 (0.01–0.02)	0.01 (0.01–0.01)	0.01 (0.01–0.01)
Western Sub-Saharan Africa	0.03% (0.02–0.04)	0.03% (0.03–0.04)	0.03% (0.03–0.04)	0.18 (0.15–0.22)	0.25 (0.21–0.31)	0.34 (0.29–0.43)

Data in parentheses are the 95% uncertainty intervals. Region and super-region numbers do not sum up the global prevalence because of rounding. YLDs, years lived with disability.

## Data Availability

The datasets generated and/or analyzed during the current study are available at https://ghdx.healthdata.org/gbd-2021/data-input-sources, accessed on 14 January 2025.

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
