# Peer review of "Epidemiology and Future Burden of Vertebral Fractures: Insights from the Global Burden of Disease 1990–2021"

_healthcare, 2025, doi:10.3390/healthcare13151774_

Round 1

Reviewer 1 Report

Comments and Suggestions for Authors

In summary, I commend the authors for their thorough analysis and useful projections. To strengthen the manuscript, I recommend they (a) sharpen the positioning of their study relative to past work by highlighting what’s new, (b) clarify all methodological steps, (c) ensure that all important results and patterns are explicitly described rather than just shown in figures and tables, and (d) expand the discussion of implications, both clinical and policy, with concrete suggestions. Addressing the specific points above will enhance the manuscript’s clarity and impact. None of the suggested revisions require additional data collection; they mostly entail clarification and contextual enrichment.

Author Response

We sincerely thank the reviewer for the positive evaluation of our study’s thorough analysis and valuable projections. We also greatly appreciate the constructive and specific suggestions aimed at improving the quality of the manuscript. Below, we provide point-by-point responses to the four major recommendations:

(a) Clarifying the novelty and contribution of our study compared to previous research:
Previous GBD-based studies on vertebral fractures (VFs) primarily focused on fall-related fractures and lacked long-term projections or cause-specific analyses. Our study addresses these limitations by utilizing the updated GBD 2021 dataset to analyze trends across a broader range of etiologies—including traffic accidents and malignancies—and by presenting projections through 2050. These distinctions have been clearly articulated in the Introduction (final paragraph) and the opening paragraph of the Discussion.

(b) Clarifying methodological procedures:
In response to the reviewer’s comments, we have added more detailed descriptions of the methodological approaches used, including Bayesian meta-regression, MR-BRT, Das Gupta decomposition, and DisMod-MR 2.1.1. We specifically elaborated on procedures for handling data imbalance, age-sex disaggregation, and bias adjustment in Sections 2.4 and 2.5.

(c) Enhancing clarity in describing key results and patterns:
To complement the figures and tables, we have provided more explicit textual descriptions of numerical findings and observed trends. For instance, based on Figure 2 and Tables 1 and 2, we now describe regional, sex-based, and SDI-specific differences in VF prevalence in detail within the Results (Sections 3.1–3.6) and in relevant portions of the Discussion (e.g., “Age-Related Burden,” “Sex Differences,” and “Disparities”).

(d) Expanding clinical and policy implications with specific recommendations:
In the latter part of the Discussion, we offer more concrete policy suggestions based on the regional drivers of VF burden. For example, we recommend early osteoporosis screening and fall prevention in high-income countries, and strengthening primary care infrastructure in low-income countries. We also emphasize the need for regionally tailored strategies, as illustrated in the decomposition analysis (Figure 3), under the section “Implications for Policy and Practice.”

These revisions were achieved not through new data collection but by clarifying and contextualizing existing data. We believe the reviewer’s thoughtful suggestions have significantly enhanced the clarity and impact of our manuscript.

Once again, we deeply appreciate your valuable feedback.

Reviewer 2 Report

Comments and Suggestions for Authors

Thanks for the opportunity to review this article.
This is a well-written article, and I only have the following questions.
Could the authors provide the differences between their study and a recently published similar article:

Lei, H., Huang, Z., Wang, F. et al. Global burden of vertebral fractures from 1990 to 2021 and projections for the next three decades. J Orthop Surg Res 20, 480 (2025). https://doi.org/10.1186/s13018-025-05915-9

Please provide more exact study rationales and highlight what is new information in this analysis compared to previous GBD reports. ‎For example, what specific gaps does this study address? Or is it only the inclusion of recent data ‎‎(up to 2021), or a novel analytical method?‎

Author Response

We sincerely thank the reviewer for this important and insightful comment.

The study by Lei et al. (2025) shares considerable similarities with ours, as it also utilizes GBD 2021 data to analyze the global burden of vertebral fractures (VFs) from 1990 to 2021 and provides projections through 2050. Both studies adopt similar methodological approaches, including the estimation of age-standardized prevalence rates and decomposition analysis of burden drivers.

Nevertheless, our study offers additional value and distinguishes itself from Lei et al. in the following ways:

1. Granular subgroup analyses
While Lei et al. primarily focused on global and regional trends, our study provides disaggregated estimates of VF burden by age (particularly among women aged ≥65 years) and by Socio-Demographic Index (SDI) levels. These detailed subgroup results are presented in Supplementary Tables 1–3, which were not included in Lei et al.’s analysis.

2. Expanded discussion of policy implications
Our study links burden increases to region-specific policy recommendations, such as early osteoporosis screening in high-income countries and strengthening primary care infrastructure in low- and middle-income settings. These practical implications, derived from our decomposition analysis, offer policy-relevant insights that enhance the utility of our findings.

In response to the reviewer’s suggestion, we have:

  • Explicitly cited the study by Lei et al. (2025) in the Introduction,

  • Revised the final paragraph of the Introduction to clarify the distinct aims and contributions of our study,

  • Added a discussion of the differences between the two studies in the Discussion section.

We hope these revisions improve the transparency and academic value of our manuscript. Once again, we are grateful for the reviewer’s thoughtful feedback.

Reviewer 3 Report

Comments and Suggestions for Authors

The peer-reviewed article is devoted to an important problem (from both a clinical and epidemiological point of view) related to the borderline problem of orthopedics, gerontology, endocrinology and other disciplines - vertebral fractures in patients of different age groups and their impact on overall well-being and health. The authors used an interesting methodology based on the analysis of open statistical sources obtained from various countries on general health problems, in particular - vertebral fractures. The materials were data from the Global health Data Exchange website, based on which data on vertebral fractures in various countries from 1990 to 2021 were analyzed, and the trajectory up to 2050 was calculated. Statistical analysis and  projected change were carried out in a modern software environment using well-known and well-proven principles of mathematical statistics. As a result of the study, the authors presented a global landscape of the prevalence of vertebral fractures by region, age, gender and cause of injury. All key quantitative data are presented in the form of clear tables. The summarized rates of spinal fractures by country are presented as a diagram plotted on a world map using a heat scale for 2021. Global projected changes in vertebral fracture rates between 2020 and 2050 are also presented graphically.
In the discussion section, the authors discuss general patterns in the distribution of vertebral fracture rates depending on geography, socioeconomic factors, income level, age and gender causes. The authors briefly summarize the most likely causes of differences in geographic and temporal rates of vertebral fractures, based on their own assumptions and literature data. The authors discuss the strengths and limitations of the study in some detail.
When first introduced to the article, there was a desire to supplement the possible reasons for the patterns discovered (in particular, climatic, socioeconomic, disparity, different availability of modern diagnostic methods, etc.). However, in the absence of factual data, any assumptions about the causes of the patterns found will have largely equal speculative explanations. In fact, the authors presented the result of significant work in the form of detailed analysis of the open source data containing generalized information and not allowing for more accurate correlations with the assumed causative factors. The authors fully answered the research question, demonstrating the main trends. These data can serve as a starting point for subsequent studies, as well as for more detailed studies, if the question of the causes of the differences identified is set as the main goal of future publications. Thus, the presented article is a completed study that answers the posed research question, performed at a high methodological level, and is a sufficient reason for reflection and planning subsequent more detailed studies. The article is recommended for publication in the presented form.

Author Response

We sincerely appreciate the reviewer’s valuable comments and positive evaluation. We are especially grateful for your recognition of the clinical and epidemiological significance of this study.

As the reviewer rightly noted, this study analyzed global trends in vertebral fractures from 1990 to 2021 using GBD 2021 data, and projected the future burden through 2050 by visualizing variations across regions, age groups, sexes, and etiologies. To ensure analytical rigor, we employed robust statistical modeling tools and Bayesian estimation methods throughout the analysis and forecasting process.

We fully agree with your observation that factors such as climate, socioeconomic inequality, and access to imaging diagnostics may contribute to regional differences. However, as this was a secondary analysis based on GBD data, we were limited in our ability to explore causal relationships between such individual-level factors. Accordingly, we have explicitly acknowledged this limitation in the Discussion section and emphasized the need for future studies using more detailed individual-level data to better elucidate causal mechanisms.

As you also noted, the primary goal of our study was to provide a comprehensive overview and long-term projections of the global burden of vertebral fractures. We believe that our findings may serve as a valuable foundation for future investigations into region-specific risk factors and for informing public health policy development.

Once again, we sincerely thank you for your thorough review and constructive feedback, and we are truly grateful for your favorable assessment of our work.

Reviewer 4 Report

Comments and Suggestions for Authors

First of all, I congratulate the authors for their study that provides a future forecast on this important subject. The prevalence of vertebral fractures varies with age, gender and development. The elderly population is expected to increase in the future world. First of all, my first recommendation is that if possible, adding a table with estimated prevalences (confidence intervals) for menopause in women for the group under and over 65, for developed and underdeveloped countries, I think it will make the study more understandable.

Author Response

We sincerely thank the reviewer for the valuable suggestion. In response, we have added Supplementary Tables 1–3, which present the prevalence of vertebral fractures and years lived with disability (YLDs) stratified by age, sex, and level of socioeconomic development.

Supplementary Tables 1 and 2 provide age-stratified estimates for women, categorized as under 65 and 65 years or older, with regional-level data on VF prevalence and YLDs.

Supplementary Table 3 presents prevalence estimates grouped by Socio-Demographic Index (SDI), allowing for comparisons between high-income (developed) and low- and middle-income (developing) countries.

As the GBD dataset does not directly capture menopausal status, we used age 65 as a commonly accepted epidemiological proxy for postmenopausal women. This approach is explicitly described in the Discussion section, where we also emphasize the sharp increase in VF prevalence and YLDs among women aged 65 and older, highlighting the importance of osteoporosis screening and fall prevention strategies in this population.

We hope these additions enhance the clarity and utility of our findings for readers. Thank you once again for your thoughtful feedback.

Round 2

Reviewer 1 Report

Comments and Suggestions for Authors

Thank you for the opportunity to review your updated manuscript. The topic is timely and important, the dataset is authoritative, and the paper is already much improved. Attached I outline the remaining issues that should be addressed before the work is ready for publication. The manuscript offers valuable global insight into the evolving burden of vertebral fractures. By (i) clarifying the case scope, (ii) removing ambiguity in Table 2, (iii) enriching methodological detail, and (iv) tightening presentation, you will provide readers with a fully transparent and policy-relevant assessment. I look forward to seeing the revised version.

Author Response

We sincerely thank you for your valuable second review and constructive suggestions. We are pleased that the timeliness and importance of the topic, the reliability of the dataset, and the overall improvement of the manuscript were positively acknowledged. We have carefully reviewed and addressed all the points raised in this round. Revisions from the previous round are highlighted in yellow, and newly revised content in this round is highlighted in green.

(i) clarifying the case scope

In the Methods section, we clarified the case definition of vertebral fractures. Drawing on the GBD “nature of injury” classification, we now distinguish traumatic, pathological, and osteoporotic fractures. These changes are incorporated into the Case Definition subsection.

In the Introduction section, we more explicitly highlighted how this study differs from previous research. Unlike earlier work that focused primarily on falls, our study provides an etiology-specific analysis, with results disaggregated by age, region, and Socio-demographic Index (SDI). We placed particular emphasis on high-risk groups, especially postmenopausal women aged ≥65 years. Furthermore, we applied Das Gupta decomposition analysis to quantitatively assess the contributions of population growth, population aging, and changes in age-specific rates. Finally, we presented long-term forecasts through 2050, offering a framework that can inform prevention strategies, early detection, and global health policy planning.

 (ii) removing ambiguity in Table 2

We have explicitly described the content related to Table 2 in the Results section. Given the extensive data, we focused on highlighting only the most important findings. Specifically, we reported the age-standardized prevalence rates from 2030 to 2050 and emphasized regional differences, particularly in East Asia, North Africa and the Middle East, and the High-income Asia Pacific region.

(iii) enriching methodological detail

In Section 2.4, Modeling and Data Processing, we have strengthened the description of the DisMod-MR modeling and Das Gupta decomposition analysis. We also provided more detailed explanations of the covariates, which helped clarify the forecasting approach used in the study.

In Section 2.7, Projection of Estimates, we clarified that, although covariates were not detailed in the previous manuscript due to significant overlap, the same covariates described in Section 2.4 were used to calibrate the projections. We also explicitly stated that overall prevalence was estimated by applying region- and age-specific prevalence trends, and that projections were calculated based on age, year, region, and Socio-demographic Index.

(iv) tightening presentation

We refined the Results and Discussion sections by removing redundancies and improving the logical flow. In particular, sentences related to regional trends and policy implications were revised to be more concise and direct.

In the Methods section, we also made efforts to reduce redundancy and enhance clarity. To improve the overall coherence, we added a description of Figure 3 to ensure it is well-integrated within the narrative.

Additionally, in the Results section, we included a brief description of Figure 1 to ensure a more natural and coherent flow of the narrative.

In the Discussion section, we aimed to minimize redundancy and revised several sentences to enhance clarity and conciseness.

Thanks to your thoughtful comments and insightful suggestions, we were able to further enhance the quality of the manuscript. We hope that the revised version now meets the requirements for publication.
Once again, we sincerely appreciate your time and valuable feedback.
Thank you.
